# Knowledge, Attitudes and Practices of Australian Trainee Childcare Educators Regarding Their Role in the Feeding Behaviours of Young Children

**DOI:** 10.3390/ijerph17103712

**Published:** 2020-05-25

**Authors:** Penelope Love, Melissa Walsh, Karen J. Campbell

**Affiliations:** 1Institute for Physical Activity and Nutrition, Deakin University, Geelong 3216, Australia; karen.campbell@deakin.edu.au; 2Centre of Research Excellence, Early Prevention of Obesity in Childhood (EPOCH), Sydney 2007, Australia; 3School of Exercise and Nutrition Sciences, Deakin University, Geelong 3216, Australia; melissa@urbanmuaythai.com

**Keywords:** childcare, childcare educators, child feeding practices, CFAPQ

## Abstract

*Background*: Early childhood (2–5 years) is acknowledged as a critical time for the establishment of healthy behaviours. The increasing number of children and amount of time spent in childcare provides strong rationale to explore the important role that childcare services and childcare educators play in influencing healthy eating behaviours of young children in their care. *Methods*: This study used a qualitative exploratory approach to describe the knowledge, attitudes and practices of Australian childcare trainee educators’ regarding their role in the feeding of young children. *Results*: All participants agreed that feeding of young children was an important part of their role, but described challenges to the promotion of healthy eating and the adoption of responsive child feeding practices. These included personal beliefs and experiences with food, the bi-directional nature of child feeding, conflicting parental requests and/or unsupportive centre-based policies and procedures. *Conclusion*: Training about responsive child feeding practices within the childcare sector should include all childcare staff; aim to enhance relational efficacy and communication skills with parents; and empower childcare staff to lead organisational change. To support this, childcare centres need to provide coherent centre-based healthy eating policies inclusive of healthy food provision and desirable feeding practices.

## 1. Introduction

Early childhood can be defined as from two to five years of age and is acknowledged as a critical time for growth and development [1,2,3]. Many factors can influence health outcomes during childhood, such as nutritional inadequacy, educational care and social interactions [4,5,6,7]. In Australia, almost one in four (21%) children aged two to five years are already overweight/obese, with those living in lower socioeconomic or regional areas more affected [8]. While the causes of overweight/obesity are associated with many factors, poor diet is a key modifiable risk factor, with eating habits and food preferences learnt during the early years having the potential to influence health over the life course [9,10].

Children are exposed to a range of environments during their childhood including the home, family, other caregivers and Early Childhood Education and Care (childcare) services, all of which play a significant role in influencing their health and development [11]. While the home and family environment are traditionally recognised as primary determinants for influencing children’s eating behaviours and food preferences [12,13], societal changes and an increase in the participation of mothers in the workforce in developed countries [14,15] have resulted in childcare services sharing this role.

The use of childcare services in Australia continues to rise [16]. These include centre-based care (long day care, occasional care and preschool/kindergarten); family day care (in approved carer’s home); home-based care (friend, relative, babysitter or nanny in own home); and outside school hours care (for primary school aged children) [17,18]. There are almost 16,000 approved childcare services in Australia, with centre-based care services comprising the majority (61%) in the form of long day care (LDC) centres [19,20]. LDC centres are defined as operating 8 h/day for at least 48 weeks/annum, and providing one main meal and two snacks per day, either prepared on-site or provided by carers [11]. In 2018, 751,450 Australian children were enrolled in LDC centres, spending an average of 25.7 h per week in care [20,21]. This suggests that LDC centres are the greatest contributors to time spent in care and provide a high level of exposure to food environments and feeding practices during an influential developmental stage.

Australian childcare services operate as accredited services through the Australian Children’s Education and Care Quality Authority (ACECQA) [22]. All services must comply with the National Quality Framework (NQF), which provides streamlined regulatory and quality standards for the industry. Assessing compliance with the NQF is undertaken at a state level by various regulatory authorities, most commonly the State’s Education Department. The NQF comprises seven national quality standards (NQS). NQS Area 2—Health and Safety includes element 2.2.1 that “healthy eating is promoted, and food and drinks provided by the service are nutritious and appropriate for each child”, with accompanying resources based on the Australian Dietary Guidelines [22].

Although comprehensive best practice standards and policies to guide childcare educators regarding food provision and child feeding practices might exist in some countries, such as New Zealand, Australia, the USA and the UK [23] and may form part of a regulatory framework, awareness and translation of “broad government recommendations” within this sector appears limited [23,24,25]. Globally, food intakes of children in LDC centres have been found to be inconsistent with dietary recommendations, for example low intakes of fruit, vegetables and wholegrains, and high intakes of saturated fat and sweet snacks [26,27,28,29]. Given the known risk of increased central and total adiposity in children with increased exposure to childcare, it is important the recommendations regarding healthy food provision and child feeding practices be adopted [30,31,32].

In addition to meeting dietary needs, exposure of children to desirable feeding practices is foundational for the early establishment of lifelong healthy eating behaviours and healthy body weight [33]. Child feeding practices are defined as “particular behavioural approaches adult caregivers employ to control what and how much children eat” [34]. Responsive child feeding is defined as a process of “reciprocity between child and caregiver” where the carer engages in positive practices to enhance self-regulation of intake, acceptance of new foods and development of healthy eating behaviours [35]. Responsive child feeding practices include allowing children to control the amount of food they eat, repeated exposures to foods, modelling healthy eating, fostering children’s interest through discussions about food and involvement with meal preparation, and providing a positive mealtime environment [34,36,37], including family-style meal service [38]. Non-responsive child feeding practices, where there is a lack of reciprocity between child and caregiver, are discouraged and include pressuring children overtly or covertly to overeat, restricting access to palatable foods, using food as a reward or bribe, and using verbal instructions based on external cues (e.g., “Have another bite” or “Are you still hungry?”) [34,39,40].

Several studies have investigated the influence of feeding practices on children’s dietary intake within the family home. A recent systematic review (*n* = 78 studies) concluded that food availability and parental modelling show the strongest associations with dietary intake; that active guidance/education might be more effective to increase healthy food intakes; and, for children six years and younger, verbal praise might be more effective in promoting healthy eating [41]. Studies have typically focused on mothers, considered to be the primary caregiver during the formative years. However, with more families reliant on childcare, the child–feeding relationship has become a shared responsibility between carer and childcare educator [13]. The influence of childcare educators on child feeding practices is therefore an important consideration.

Childcare educators are important role models given the length of time children spend in care, and can aid the development of a child through play based learning, social interaction, school readiness, and food practices [30,42,43], as well as enhance behaviours learned at home [23]. Beyond modelling, a childcare educator’s unique personal values, beliefs and perceptions, age, education level, ethnicity, health status, length of employment in childcare and training can also impact their behaviours and practices with children in their care [14,34,44,45,46,47,48,49]. While Australian childcare educators are required to have a minimum of a Certificate lll in Early Childhood Education and Care [22], Australian studies [50,51] show that childcare educator nutrition knowledge varies greatly, and personal experiences mostly shape attitudes and practices [52]. The majority of childcare educators however perceive themselves as having a high standard of nutrition knowledge [50,51].

The increasing number of children and amount of time spent in childcare provides strong rationale to explore the important role that childcare services and childcare educators can play in providing supportive food environments and feeding practices to young children in their care [24,53,54,55]. A focus on the provision of healthy environments during the early years is also considered the most cost effective obesity prevention strategy [56]. In Australia, studies exploring the knowledge, attitudes, beliefs and practices of childcare educators regarding the feeding of young children in childcare are scarce [50,51], and no studies exist regarding trainees still undertaking entry-level qualifications. This study is an exploration of the knowledge, attitudes and practices of Australian childcare trainee educators’ regarding their role in the feeding of young children to provide insights in how best to support this workforce in the adoption of responsive child feeding practices.

## 2. Materials and Methods

This study used a qualitative exploratory approach underpinned by a contextualist epistemology, where knowledge emerges from and is situated within the contexts of the data [57]. This was a pragmatic and appropriate approach given the novelty of the research question, with the intent of exploring constructs to inform future examinations of the area. The roles and backgrounds of the researchers are made explicit given that data interpretation can be influenced by these. All researchers are mothers and qualified dietitians/nutritionists working within the field of nutrition in the early years (infants/children under 5 years). Ethical approval for this study was obtained through Deakin University (HEAG-H 143_2018). The researchers followed the Consolidated criteria for reporting qualitative studies (COREQ) checklist [58].

### 2.1. Recruitment of Participants

In Australia, childcare educators undertake either a Diploma of Early Childhood Education and Care, or a Certificate lll in Early Childhood Education and Care [22], which are provided through registered Technical and Further Education (TAFE) training institutes. TAFE courses are designed as a pathway to undergraduate degree courses offered through universities, and all courses meet national educational standards. The Certificate lll in Early Childhood Education and Care is a full-time (3 days/week) course over 12 months, including 210 h of “work placement” across three different childcare centres. The certificate includes 15 core and 3 elective units of study and is a pathway to the Diploma in Early Childhood Education and Care. The diploma is a full time (3 days/week) course over 18 months, including 360 h of “work placement” across three different childcare centres. The diploma includes 23 core and 5 elective units of study and is a pathway to a Bachelor of Education degree. Both the certificate and diploma include the unit “Promote and provide healthy food and drinks”. While on “work placement” students are involved in the daily routines of the centre. Within the childcare setting, certificate graduates operate as “room assistants” and diploma graduates as “room leaders/teachers” with opportunity for more senior management roles.

Convenience sampling was used to recruit participants from a Victorian TAFE institution offering childcare educator training at its regional and metro campuses. Organisational consent was obtained prior to approaching enrolled students at each campus. A short presentation about the study was made to students enrolled in the Diploma of Early Childhood Education and Care and the Certificate lll in Early Childhood Education and Care. The presentation was delivered by P.L. at a convenient time, arranged by the campus coordinators, while students were onsite attending usual classes. This presentation outlined the importance and purpose of the study, and what would be expected of study participants. Interested students provided a contact email, were directed to a 5-min online survey to collect personal demographic data (Diploma/Certificate enrolment, age, highest educational level, home language, ethnicity, sex, and number and age of own children), and offered the opportunity to participate in a 60-min on-campus focus group.

### 2.2. Data Collection—Focus Groups

Participants self-selected which focus group to attend. Consent to participation and to audio recording were reconfirmed at the start of each focus group. Fruit platters were provided during sessions and all participants received a coffee voucher at the end of their session. As an experienced qualitative researcher and focus group facilitator, P.L. facilitated all focus groups. One observer attended each session to take notes. Observers were students undertaking postgraduate nutrition studies.

Discussions were facilitated using a focus group topic guide and photo elicitation. The focus group topic guide contained questions and prompts as described in the Childcare Feeding and Activity Practices Questionnaire (CFAPQ) [37]. The CFAPQ is a tool to assess childcare educator food and physical activity related practices [37], comprising sixty-three items (40 food-related items and 23 physical activity-related items). Only child feeding practices were explored in this study informed by the food-related items—restriction (6 items), monitoring (4 items), modelling/encouraging balance and variety (7 items), involvement/environment (5 items), teaching (3 items), pressure to eat (4 items), child control (5 items), emotion regulation/food as reward (5 items) and healthy foods offered first (1 item) [37]. This tool was not administered as a survey but was used to prompt questions during the focus groups. Photo elicitation [59,60], in the form of four photographs depicting various child feeding practices in a childcare setting, was used in conjunction with the topic guide to provide visual stimulation and generate in-depth conversations. (File S1: Interview questions, prompts and photographs).

### 2.3. Data Analysis

Demographic data via online survey responses were analysed using descriptive statistics. Focus group audio-recordings were professionally transcribed verbatim. Transcripts were inductively coded by M.W. using NVIVO v12 (QSR International, Melbourne, Australia), with coding verification by P.L. NVIVO coding summaries were used for content analysis to identify themes which were explored in relation to the CFAPQ feeding practice domains [61]. Consensus on final themes was developed in agreement among all authors. As an exploratory study with a small sample size, data saturation was not a consideration [57].

## 3. Results

Fifty students (Campus 1 *n* = 30, Campus 2 *n* = 20) attended the study recruitment presentation. Nineteen students (Campus 1 *n* = 9; Campus 2 *n* = 10) consented to participating in the focus groups, with each student participating in one focus group. Three focus groups, with 5–9 students, were held.

### 3.1. Description of Participants

All study participants were female ranging in age from 18 to 59 years, with a median age of 27.7 years. The majority (84%) of participants were Australian born, with three participants born overseas (China, Burma and the United Kingdom). All participants were fluent in English. The majority of participants (74%) had completed Year 12. Over half (58%) were enrolled in the Diploma in Childhood Education and Care, 42% in the Certificate lll in Childhood Education and Care, and all had completed some “work placement” hours. Participants with children of their own (47%; *n* = 8) had mostly young children (2 months–11 years of age) and were from the same TAFE campus (Table 1).

### 3.2. Trainee Childcare Educator Perspectives on Feeding Young Children

#### 3.2.1. Childcare Educator Role

The majority of participants (*n* = 11) described the importance of their future role as a childcare educator in terms of *“being a positive influence”* and *“nurturing children”* to *“help shape their future”* from an educational perspective.


*“I’m training to be an early childhood educator because I believe that the profession is leaning towards education rather than just childminding”*
{G7; age 59 years; parent}

Some participants described the childcare educator role as that of a “proxy mother”.


*“I know what it’s like to leave your kids at childcare, so I want to be like a second mum to the children so they (carers) know that they can drop their kids off and feel safe”*
{2W1; age 35 years; parent}

When reflecting on their own practices as a parent, however, participants described the importance of following best practice within the childcare centre rather than defaulting to their own parenting style.


*“I know as a parent, I’ve done that to my children before, tried to shove food into their mouths. But as an educator I’d never do that. So yeah, I think an educator’s role is different”*
{1W3; age 32 years; parent}

Regarding the feeding of young children in their care, all participants agreed that this was an important part of the childcare educator’s role. Their concerns about childhood obesity and the ubiquitous presence of “sweets and fast foods” appeared to inform this opinion, with the promotion of healthy eating at an early age therefore regarded as essential.


*“Because child obesity is becoming a big thing now and children do spend the majority of their time in our care, rather than their parents”*
{2W3; age 31 years; parent}


*“I think definitely a part of an educator’s role is to introduce those healthy habits so they become habits throughout the rest of the children’s lives”*
{G6; age 18 years}

Participants described a number of challenges to the promotion of healthy eating and child feeding practices, with (social) media, marketing and parents regarded as *“strong influences”*.


*“A big challenge is what they (parents) see through media … whether it’s social media or whether it’s just seeing things pop up when they’re watching YouTube … ads for fast food and sweets and lollies… it’s hard to try and get past that when it’s such a strong influence now”*
{G3; age 18 years}

Being a positive role model to children and their families was therefore considered an important, although challenging, aspect of the childcare educator role. Speaking directly with parents and/or via newsletters and emails were described as ways to communicate about healthy food choices, packed lunch options, and feeding practices that could be used at home.


*“I believe that it could be quite difficult speaking with some parents about what you do not consider to be a good food choice…, trying to explain to them without upsetting them or making them feel judged.”*
{G9; age 18 years}

#### 3.2.2. Child Feeding Practices

Content analysis of focus groups identified all child feeding practice domains within the Childcare Feeding and Activity Practices Questionnaire (CFAPQ) and three emerging domains: mealtime environment, policy environment and home environment (Table 2).

##### Restriction—Guidance on Food Regulation (Type and Amount)

All participants considered it important to limit unhealthy food provision in the childcare setting, including food provided from home. Participants felt that unhealthy food provision from home occurred due to parental confusion around healthy food options and a perception that homemade foods were healthier than processed, packaged options. Some participants suggested framing food provision as an issue of “equity” to encourage parents to provide similar, healthy options.


*“…one child’s lunch, it was like a packet of chips and biscuits, and so the educator spoke to the mum and said, you know, “We can’t really be having this food because it’s not healthy and it’s not fair on all the other children that one child gets to eat a packet of chips and the others are having fruit and vegetables.”*
{2W7; age 20 years}

Birthdays were an extra challenge for childcare services, and restrictions were common for the types of “celebration foods” allowed. Some childcare services were described as making celebratory food on-site to ensure it would meet the needs of all children regarding allergies. Other services held one celebration for all birthdays within the month, with parents providing small cakes so “the children have a little piece each” or a small cake “for the blowing of the candle” but this cake went home to the family.


*“because you know there would be hundreds of cakes every day.. so when there is a kids’ birthday, they do a small cake to celebrate the birthday for the kid, but they never give the cake out to anyone to have it, they give it to the parents to take back home”*
{2W1; age 35 years; parent}

##### Monitoring—Tracking of Foods and Drinks Consumed

Some participants described observing daily monitoring of each child’s food and drink intakes as routine childcare procedure to provide feedback to carers. Participants felt that this information was useful for carers to gauge their child’s food intake; however, it also posed challenges when carers felt their child was eating too little or too much.


*“.. a few times we had one parent say “He needs more than one, he needs to have two, three servings” because maybe he’s gone home hungry or something.. so then we’ll try and give them that second one, we’ll ask them first “Do you want another one?” or if they’ve scoffed it down then we know that they would like another one”*
{1W1; age 30 years; parent}

##### Modelling/Encouragement—Role Modelling Healthy Eating, Food Enjoyment

The influence of modelling was described as role modelling (performed by the adult carer) and peer modelling (performed by the child/ren). All participants expressed the belief that a childcare educator should sit with children and eat the same food during mealtimes, to provide appropriate learning opportunities and encouragement of eating new foods. Participants however described feeding large numbers of children as a barrier to this, and had observed a mix of role modelling behaviours with some childcare educators sitting and eating with children, some sitting but not eating with children, and some just “walking about” overseeing mealtimes.


*“How do you expect the children to want to eat that ‘not so appealing looking’ food, when the educator isn’t herself?”*
{G5; age 19 years}

Participants shared the opinion that children who frequently refused foods at mealtimes in childcare were likely to be eating a limited variety at home.


*“So if you’re trying to serve children at the centre broccoli and vegetables and they’re not wanting it, it could be because they’ve never had anything to do with the food. I think it’s important to discuss with the parents or the guardians maybe starting to serve it at home and role model it at home as well as the centre. That way it becomes regular and normal”*
{G3; age 18 years}

Participants also recognised the importance of peer modelling and the role of group mealtimes to enable the mimicking of feeding behaviours amongst children. Participants described observing childcare educators encouraging children to share with each other about why they like a food, and commented on how carers frequently remarked that children ate foods at childcare that they did not eat at home.


*“So if they see everyone else doing it they might be a little bit more inclined to try it, realise they like it, and now they’re having lunch”*
{G5; age 19 years}

##### Involvement—Assistance with Meal Preparation/Mealtimes; Variety and Balance on Offer

Involvement was described in terms of children helping to prepare meals, serving of food and tidying up at completion of mealtime.


*“At one centre they cleaned off their own plates into a food scrap bin and put plates into a pile and cutlery into a water container… I like the idea of bringing the children out to help them set the tables. They did help pour their own glasses of water, so little jugs. So those self-help skills were present.”*
{1W2; age 37 years; parent}

Participants expressed different views about how much involvement the childcare educator should provide, with some concerned that if they helped too much this could affect a child’s choice and self-regulation. For some participants, learning manners and eating with cutlery were seen as more important than involving children in helping with the serving and carrying of food. While the involvement of children in mealtime preparation appeared limited, some positive observations were shared by participants.


*“At my centre recently they made healthy mini pizzas, they got little muffins and they had a table set up with mushrooms and spinach and tomato and the children could get the tongs and make it themselves”*
{2W3; age 31 years; parent}

Participants also discussed the importance of providing colourful, appealing meals and snacks, and how this might be aided by having a vegetable garden within a childcare centre. Growing and preparing the harvested produce was described as a positive way to involve children in mealtime preparations.


*“I think it’s really important to include the growing of healthy foods in your centre. That way when children contribute to what they see in front of them it really makes it a bit more exciting for them and it’s just more incentive for them to eat it. They know where it’s come from”*
{G9; age 18 years}


*“… and also perhaps children could be shown a recipe.. that if you put it all in together in a vegetable soup they start to like it..”*
{G7; age 59 years; parent}

##### Teaching about Nutrition—Discussion about Healthy Eating, Foods, Nutrient Value

Participants described the promotion of healthy eating to children and their carers as an important role for a childcare educator, such as having conversations with children during mealtimes, and talking about the types and benefits of the foods on offer. Apart from mealtimes, some participants described providing resources for children to develop their food literacy through role-play.


*“if you regularly explain to the children how important it is for you to eat these [foods] they might be inclined to eat them..”*
{G5; age 19 years}


*“I believe it [play equipment] should be included in the centre, like a little kitchen area for them with all the plastic foods that are healthy so they can just become more familiar with these things and talk about them”*
{G9; age 18 years}

##### Pressure to Eat—Coercion to Eat More

Most participants believed there was no need to force a child to eat, however some expressed concerns about children who might say they were full or not hungry because they disliked the food provided. Tension between respecting the child’s decision and meeting the child’s nutritional needs was evident.


*“You have to respect children’s choices to an extent…but a lot of the time when a kid says they’re full they might not be, they might say that because they don’t want to eat it…If a kid’s always not eating then you need to take steps to work towards getting them to eat regularly. So, I think it’s a balance between the health and safety of the child and also respecting their wishes…”*
{G6; age 18 years}

Passive coercion was also discussed by participants, such as hiding frequently disliked foods, such as vegetables, in mixed meals. Some participants commented how “*children are smarter than you think and will know and just not eat it*”, and suggested alternate flavourings and preparation methods with an explanation about “*why they need to eat this food and why it’s good for them*” {G5; age 19 years}.

##### Child Control—Choice, Alternative Options, Self-regulation

Child control was described in terms of decision-making and choice, and all participants agreed that children were the best moderators of their own satiety. Participants believed that a child’s decision-making about mealtime (what foods to eat and how much) was based on several connected factors: feelings of hunger, appeal of food being provided, behaviour of peers, mealtime environment and encouragement from childcare educators.

While all participants agreed that a child’s decision to eat or not should be respected, there were varying opinions about whether children should self-serve (thereby deciding which foods to put on their plate) or be served (thereby deciding how much to eat of all the foods on the plate). Concerns about self-service included nutritional inadequacy (with children selecting only the foods they like) and mealtimes being messy. Pre-served meals were associated with wastage, disrespecting a child’s autonomy and the likelihood of large portions being encouraged. Role modelling was considered pivotal in helping guide children in the selection of both type and quantity of food whether self-served or pre-served.


*“I think it’s important that educators understand that when teaching young children the importance of self-serving, it’s going to be messy because children are adapting possibly to different situations than there are at home”*
{G3; age 18 years}


*“They’re all being served the same amount, so they probably won’t all eat that. I think sometimes that influences bad habits too like “Eat everything on your plate”*
{1W7; age 28 years; parent}

##### Emotion Regulation—Food When Fussy, Unhappy; Reward; Punishment

Participants described observing child emotions in relation to food provision such as happiness, excitement, discontent, discomfort and disgust. In response to this, participants observed warnings of rest time or going hungry, with the most common childcare procedure being the provision of an alternate food choice such as fruit, a plain main meal or a plain sandwich. Providing different options however appeared to pose an issue when other children saw this and requested these as well, and caused conflict with carers where children expected this practice to occur at home.

For some participants, a tension existed regarding fussy eating and coaxing a child to try a new food, where words of encouragement and role/peer modelling could be considered coercion.


*“..like with fussy eaters… the educator would say “Just try a little first”, we would encourage them, even just to have a spoonful…. before getting the chef to make more food”*
{2W7; age 20 years}


*“If a child doesn’t eat what they’re given, they try and get the other kids to encourage them, you know, “It’s really yummy..”*
{1W3; age 32 years; parent}

##### Mealtime Environment—Progressive Meals

Participants regarded mealtimes as an important time for children to develop social skills and suggested mealtime ambiance and environment played a central role to the success of mealtimes in childcare centres. Relaxed, semi-structured mealtimes of small groups were described as enhancing desirable child feeding practices. Some participants had observed lunch as a “progressive” meal where children could self-nominate when to eat during the allocated one-hour lunchtime. Participants felt this resonated well with respecting a child’s autonomy and self-regulation as well as considering that some children feel uncomfortable in large groups.


*“..progressive lunches are a really good thing because you ask a child at the start [of lunch].. if they say they’re not hungry or they do get to the table and they don’t feel they’re ready you can ask them to come back in another group. It gives them time to think about it, see other children eating”*
{G3; age 18y)

##### Policy Environment

All participants agreed that while on placement they had not observed all the desirable child feeding practices they were being taught. Most were unsure of what information was provided to parents regarding policies and procedures about healthy food provision at childcare centres.


*“During my placement, when you go into policies and procedures it’s in there in the Q&As. But unless you’re looking for it, you don’t see it. So parents don’t really know upfront”*
{1W1; age 30; parent}

##### Home Environment

While all participants felt they understood why healthy eating was important in childhood and had the content knowledge, many expressed concerns about how to extend the application of this information to the home environment. Participants suggested that providing carers with healthy eating information (such as recipes and newsletters) was important to complement what the child was experiencing at childcare.

*“I think it’s more than just being in your comfort zone as such. It’s more just looking out for the best interests of the children and because of that I think it’s your duty to go and speak to a higher power if you have concerns about the children’s current nutrition”*.{G9; age 18 years}

Most participants stated that as childcare educators their role was different to carers. Participants perceived parents as placing more pressure on getting children to eat at mealtimes, compared with childcare where they were striving to have mealtimes that were calm, encouraging and centred around teaching children healthy feeding practices.


*“..in child care, we encourage them to feed themselves, give them time, socialise. It’s a bit different, because at home it’s like, just finish the food first, you know. We don’t have all day.”*
{2W6; age 25 years; parent}

#### 3.2.3. Childcare Educator Nutrition Training

Participants felt there was a gap between knowledge learnt at TAFE about child feeding practices and what was observed when on placement.


*“we learn a lot of things here and then get into the workplace and some of it is questioned and some of it definitely doesn’t follow what we learn at TAFE”*
{G3, age 18 years}

Participants felt they had gained new knowledge regarding desirable child feeding practices, especially self-regulation and not pressuring children to eat.


*“At home my daughter would eat really slow, so I sometimes forced her to just finish the food…we don’t have all day, which is different to what we encourage for childcare”*
{2W6, age 25 years; parent}


*“After going into the course it’s kind of taught me that they (children) know what they need if you teach them. So as long as you educate them in the right direction they’ll be able to make those responsible decisions”*
{G5, age 19 years}

Participants who were also parents expressed that they could now “see it from both sides” when trying to implement healthy eating practices in the childcare setting. Some participants described applying their child feeding knowledge with their own children at home.


*“One thing that I’ve done with my daughter is when she’s eating say carrots, I’ll sit there and talk to her about how they’re good for her eyes… so just thinking about the food in a positive way. And at the supermarket I got her to pick the broccoli out for dinner and then when we’re eating dinner, she’s proudly telling my husband”*
{1W3, age 32 years; parent}

## 4. Discussion

The aim of this study was to describe the knowledge, attitudes and practices of Australian childcare trainee educators’ regarding their role in the feeding of young children. Our study reveals that a range of individual and contextual factors may modulate the implementation of evidence-based child feeding practices within the childcare setting.

Individual level factors that influence best practice child feeding in the childcare sector are described in the literature as including perceptions of the childcare educator role, childhood experiences (especially regarding food insecurity), personal knowledge, beliefs and attitudes and ethnicity [14,34,44,45,48]. While studies report that childcare educators perceive the promotion of healthy eating positively [24], preparing children for school, rather than developing healthy habits, is considered to be their primary role [46]. Within our study, participants also described their role as future childcare educators mostly from an educational perspective, but agreed that modelling and promoting healthy eating was an important part of this role given the opportunity to establish healthy eating behaviours early in life. Although our study participants alluded to childhood obesity as a reason to promote the early establishment of healthy eating behaviours, adjusting child feeding practices as a means of addressing body weight was not raised. Other qualitative studies however have reported a greater use of controlling feeding practices where educators are concerned with a child’s weight status [34,40], and lower levels of self-efficacy to model healthy eating if concerned with their own weight status [44].

Our study participants regarded the implementation of best practice child feeding within the childcare setting as essential, and all participants felt they had gained new knowledge through their training, particularly about the importance of encouraging child control and self-regulation around food. Participants who were parents acknowledged that applying these guidelines at home with their own children was challenging, but still important. This alignment of personal belief and best practice guidance is of interest as studies usually report a misalignment between these for educators, with personal beliefs taking precedence over best practice [24,40].

A key finding in our study was the existence of a gap between learnt knowledge and observed practice, described across all focus groups. The use of controlling child feeding practices appeared to be most commonly observed, including restriction, monitoring, emotion regulation, and pressure to eat. Peer modelling was the only responsive feeding practice commonly observed by participants whilst on placement at childcare centres. It is not uncommon for educator child feeding practices to be reported as sub-optimal [25]. When comparing observational versus self-report studies, researchers emphasise that all child feeding practices are unlikely to occur within one mealtime, therefore several time points should be studied to accommodate variations in feeding practices across the day or week [13]. Desirable feeding practices are also more likely to be recorded in self-report studies due to social desirability bias [55]. In our study, participants perceived their limited observation of desirable child feeding practices within the childcare setting to be the result of parental influence, established childcare procedures and training of centre staff.

Communicating with carers about food and nutrition was considered a major challenge by our study participants, with perception that carers are “confused” about healthy food/drink choices and make dietary requests to restrict certain foods (for example, gluten-free) with no medical basis. Participants felt that parental concern about the volume of food eaten whilst in care encouraged the use of controlling child feeding practices within the centre. Participants were also of the opinion that child food refusal was linked to limited food variety provided at home and were concerned about how to extend information about healthy eating to the home environment.

Most studies report low levels of confidence among childcare educators to approach parents about their child’s eating behaviours, especially when confronted with parental resistance [62], with a tendency to discuss general food intakes rather than specific eating behaviours [52]. Childcare is a customer-driven business therefore childcare educators have a high respect for the parent’s role and can feel uncertain about how to address parents in a non-judgmental manner if they disagree with parental practices [46]. This situation is exacerbated further when a tension exists between parental and childcare educator perceptions of their roles regarding child feeding. Parents commonly perceive the role of childcare educators as subordinate [63], and childcare educators report frustration that parents undervalue their role and see it as being a “babysitter” [46]. Childcare educators may perceive their own nutrition knowledge to be superior to parents [51], and assume parents are providing an unhealthy eating environment at home that undermines healthy eating practices implemented at childcare [52]. The accuracy of a childcare educator’s nutrition knowledge however appears to vary based on personal interests and access to relevant resources [24,46].

Parents are traditionally considered to be decision-makers for their child/ren; however, it is not unrealistic for childcare educators to assume the role of “proxy parent” for children in their care. The challenge occurs when parental and childcare educator views about the care of the child are in opposition, which can frequently occur when discussing eating behaviours. The traditional approach in dealing with such situations has been the “best interests” principle, where the course of action taken is in the best interest of the child [64]. This approach can be problematic, however, as interpretation of “best interests” is subjective. The “do no harm” principle is therefore becoming a preferred approach, where the course of action taken is to ensure the child comes to no harm [64]. What constitutes harm therefore needs to be clearly articulated in relation to child feeding, and included in childcare policies and information provided to parents and carers.

Sisson et al. (2017) [46] reported that communicating with parents is essential for childcare educator motivation; that educators feel discouraged if their healthy eating efforts are unsupported within the home environment. Working with childcare educators to increase their efficacy in having mutually respectful, constructive conversations with families about potentially sensitive feeding issues is therefore an important consideration if learnt knowledge is expected to translate into practice [24,46,55,62].

The childcare environment was regarded by our study participants as another limitation to the implementation of responsive child feeding practices, specifically current mealtime practices with large numbers of children being fed at the same time. Participants remarked that this appeared to limit involvement of children in mealtime preparations, and was not conducive to role- or peer- modelling, nutrition education or child self-regulation. Harte et al. (2019) [65] described mealtimes at childcare as a “unique cultural phenomenon”, with important routines and rituals that can be used by childcare educators to support the development of healthy eating behaviours and food preferences.

Family style meal service (FSMS), where children are allowed to serve themselves and select their own portions from communally offered foods and drinks, is considered best practice for the implementation of responsive child feeding within the childcare setting [36,38]. FSMS encourages child involvement and child control, benefits the development of a child’s gross and fine motor skills and social skills, provides a calmer environment and results in less food wastage [38,40]. When childcare educators sit and eat with children, FSMS also facilitates role modelling (educator–child) [36], peer modelling (child–child) [39,66] and opportunities to provide nutrition education [38,46]. Child involvement at mealtimes further adds to their eating enjoyment, increasing their self-efficacy for selecting healthy foods [39].

Childcare services, however, appear cautious to adopt FSMS, citing time constraints to the scheduling of seated, slow-paced mealtimes [23,39]; low ratios of childcare staff to children (1:11 in some cases) [52]; perceptions that FSMS is resource intensive and creates a messy, unhygienic eating environment [38]; a belief that children are too young to prepare or select their own food [39]; and a belief that children cannot self-regulate and will self-serve inappropriate portion sizes [38]. While our study participants agreed that child self-regulation was important, they appeared conflicted about how centres could achieve policy requirements to meet the child’s dietary needs whilst in care if children were allowed to determine their own portion sizes through self-service (FSMS). Dev et al. (2014) [38] encountered this same dilemma, suggesting that a balance be achieved with childcare services taking responsibility for providing age-appropriate portion sizes of foods during mealtimes, and children being responsible for feeding themselves and determining how much to eat.

Uncertainty about managing child food refusal appeared to create a tension for our study participants, between the desire to role model enthusiasm and encouragement around eating versus pressure to eat, and the interplay of this with the concept of child control and self-regulation. Participants described the primary drivers for the use of controlling feeding practices in these circumstances as being a concern about meeting the dietary needs of the child and a fear of parental criticism if carers perceived their children went hungry while in care. The literature suggests that childcare educators adjust their feeding practices in response to child food refusal (verbal and non-verbal), using controlling feeding practices (coercion, threats, insistence and spoon-feeding) [42,54] and the provision of alternatives (rather than repeated exposure of the target food) [52] appears to be common practice. When reflecting on the efficacy of a childcare educator to respond to child food refusal, Wolstenholme et al. (2020) [67] described a carer’s beliefs about attributions for fussy eating behaviour and hunger regulation to be highly influential. Carers who are able to determine if food refusal is due to sensory sensitivity or neophobia and who support self-regulation are less likely to regard children as fussy eaters and more likely to utilise responsive child feeding practices [67]. These authors also described relational efficacy (beliefs and confidence in another person’s estimation of one’s own ability) as important as self-efficacy (beliefs and confidence in one’s own ability) when upskilling carers about child feeding.

Understanding and supporting child control and self-regulation appear key to a childcare educator’s ability to implement responsive child feeding practices [42]. It is understood that personal beliefs, such as concerns about own health and weight status [39], past and current food security [47] and cultural dietary practices [55], all influence a childcare educator’s child feeding practices. New practices however can be taught by acknowledging existing educator beliefs and experiences [48] and providing specific knowledge on how to apply a particular practice [49]. In the case of child control, this could involve verbal examples to operationalise responsive feeding practices, such as trialled by Dev et al. [54] using “This pineapple tastes so sweet and juicy. Would you like to try it?” rather than “Be brave and try some pineapple”.

It is interesting to note that few participants in our study were made aware of centre-based policies and procedures pertaining to the feeding of children in care. Those who had observed these regarded them as being in conflict with the implementation of responsive child feeding practices, such as the provision of alternative food choices for fussy eaters, recording child food intakes for parental feedback, not providing meals for childcare educators and/or not encouraging educators to sit and eat with children. Our study participants were also concerned about parental awareness of childcare centre policies and procedures regarding the feeding of children in care.

The existence and implementation of healthy eating and physical activity policies within the Australian childcare sector is reported as low [68]. In New Zealand, Gerritsen et al. [69] reported a third of services in their study with no written policies for foods/drinks from home, making it difficult for parents to know what was expected, and, where services had policies, these were vague with staff struggling to operationalise them. The existence of centre-based healthy eating policies can provide a platform for childcare staff to frame their practices and advice to parents in the context of regulatory requirements, and to position the centre as a credible provider with standards in place [63]. An issue arises, however, when centre-based policies differ from each other in their interpretation of state or national dietary guidelines. A common example is the way in which centres manage unhealthy foods/drinks provided from home, with some centres restricting them completely and sending them back home at the end of the day and other centres allowing the child to eat unhealthy foods after eating healthier foods [51].

Where centre-based healthy eating policies do exist, they focus on the types and quantities of foods/drinks allowed whilst in care, with little/no mention of child feeding practices. Providing clear guidance within centre-based policies about desirable child feeding practices provides potential opportunities for knowledge and skill sharing between childcare educators and carers [13,34]. Centre-based healthy eating policies can also ensure the alignment of other centre activities, as described by Wallace et al. [52], such as healthy fundraising (restricting the selling of unhealthy foods/drinks), appropriate food-related activities (vegetable gardening rather than the baking of cupcakes) [69], and providing play equipment that prioritises planned learning experiences and role-play about healthy eating (such as a play kitchen/shop and food models).

Training and professional development was the third limitation that our study participants described in relation to the implementation of responsive child feeding practices. Participants felt that their training had provided them with the knowledge of best practice child feeding. Many participants raised concerns about how to discuss changes within the childcare centre when they would be the staff member with the least experience, such as suggesting menu changes to the cook or recommending FSMS to the centre director.

To create and maintain a healthy food environment within the childcare setting, a culture must exist where child nutrition and health-promoting behaviours are prioritised, and appropriate childcare staff training is considered a prerequisite in creating this culture. Gerritsen et al. (2019) [70] described childcare staff training as an opportunity to not only increase knowledge and skill development but also to facilitate a change in attitude, habits and self-efficacy to create the prerequisite motivation and enthusiasm to be positive role models. Although there is limited research on the effect of nutrition education with childcare staff, studies within Head Start childcare centres in the USA have shown significant associations between improved self-efficacy and increased knowledge of health-promoting behaviours [62] and a study of Irish preschools found nutrition education of staff resulted in considerably healthier nutrition environments [71].

Nutrition training of childcare centre staff is therefore essential, including cooks, food preparation staff, educators and directors, given their roles in influencing food provision, mealtime environments and feeding behaviours. In Australia, cooks and food preparation staff in childcare centres only need a food safety certificate or equivalent, with no nutrition training, yet they are required to compile nutritionally adequate menus with adjustments to meet food allergies and intolerances, cultural preferences and personal requests [72]. In addition to nutrition knowledge, cooks and food preparation staff should also receive training in how to prepare and serve foods in appealing ways to support the implementation of healthy feeding practices [27]. Australian childcare educators receive one unit of nutrition information within their 12–18 month courses, but recall of nutrition concepts or child feeding practices appears low, with a reliance on newspapers, magazines and food company brochures for nutrition information [52]. Considering this reliance on personal nutrition knowledge, training should provide clear and consistent guidance and practical strategies about healthy eating and desirable child feeding practices [72]. Training should also reiterate the important, often undervalued, role of the childcare educator in shaping the health of children within their care [46]. An important component of training should include communication skills to enhance collaborative parental engagement [46,63,72], to teach children about nutrition [55] and to create champions to drive change within the childcare environment [36]. Without the necessary knowledge, confidence and skills to understand and enact health promotion practices, it is unlikely that changes to enhance healthy eating behaviours and food provision within the childcare setting will be implemented by childcare staff.

## 5. Strengths and Limitations

Our study used a convenience sample who self-selected to participate in the focus groups. Study participants may therefore not be representative of all trainee childcare educators and may possibly represent those educators most interested in the area. While our sample size was consequently small, reducing generalisability of study findings, this was not the intent as the qualitative research approach is to explore the lived experience and related contextual factors. There are limited studies in Australia about the child feeding knowledge, attitudes and practices of childcare educators, and no studies exploring this topic amongst trainee childcare educators. This study therefore adds to the body of evidence regarding the needs of childcare educators, in particular, insights about training and curriculum development, to empower them to be champions of change within the childcare setting. To inform our qualitative focus group questions, our study used the Child Care Food and Activity Practices Questionnaire, which has been specifically designed for the childcare setting rather than the home environment.

## 6. Conclusions

Childcare is a critical environment where young children can develop positive lifelong food habits, and childcare educators have the potential to play an important role in influencing the health and development of children in care. Our trainee childcare educators showed an awareness and support for the use of responsive child feeding practices, especially the importance of nurturing child control and self-regulation. While knowledge of and attitudes to child feeding were aligned with current evidence, our participants described a lack of confidence communicating with parents about their child’s eating behaviours. Childcare educator training about desirable child feeding practices should therefore aim to enhance relational efficacy and communication skills with parents, as well as empower childcare staff to lead organisational change. For newly qualifying childcare educators, who are now receiving training on the importance of responsive child feeding practices, implementation is likely to be limited if childcare centres lack supportive centre-based policies and practices. To support childcare educators in their endeavours to influence the development of healthy eating behaviours of young children in their care, childcare centres need to have a coherent centre-based healthy eating policy that enables the provision of healthy food and desirable feeding practices by all staff and carers.

## Figures and Tables

**Table 1 ijerph-17-03712-t001:** Demographic characteristics of focus group participants.

Characteristics	Campus #1		Campus #2		Total	
	*n*	%	*n*	%	*n*	%
Female	9	100	10	100	19	100
Age						
18–4 years	7	78	2	20	9	47
25–30 years	0	0	3	30	3	16
31–40 years	1	11	5	50	6	32
>40 years	1	11	0	0	1	5
Education						
Year 9	1	11	0	0	1	5
Year 10	2	22	2	20	4	21
Year 12	3	34	5	50	8	42
Diploma/Certificate	2	22	1	10	3	16
Degree	1	11	2	20	3	16
Own children						
None	8	89	2	20	10	53
1 child	0	0	3	30	3	16
2 children	1	11	4	40	5	26
>2 children	0	0	1	10	1	5
TAFE enrolment						
Certificate III	5	55	3	30	8	42
Diploma	4	45	7	70	11	58

**Table 2 ijerph-17-03712-t002:** Focus group themes mapped to CFAPQ and emerging domains.

CFAPQ Feeding Practice Domains	Focus Group Themes
Restriction	Guidance on food regulation (type and amount)
Monitoring	Tracking of foods and drinks consumed
Modelling/Encouragement	Role modelling healthy eating, food enjoyment
Involvement	Assistance with meal preparation and mealtimes; variety and balance of healthy foods on offer
Teaching about nutrition	Discussion about healthy eating, foods, nutrient value
Pressure to eat	Coercion to eat more
Child control	Choice, alternative options, self-regulation
Emotion regulation	Food when fussy, unhappy; reward; punishment
Emerging domains	
Mealtime environment	Structure of mealtimes and surroundings
Policy environment	Alignment of content learnt and practice observed
Home environment	Communicating with families

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
