# Peer review of "Knowledge, Attitudes and Practices of Australian Trainee Childcare Educators Regarding Their Role in the Feeding Behaviours of Young Children"

_ijerph, 2020, doi:10.3390/ijerph17103712_

Round 1

Reviewer 1 Report

This is an important topic that has global application.

Lines 28-62 Well-articulated presentation of the child care feeding situation in Australia.  Thank you for laying a foundational base for the study.

Lines 74-74 Please consider other definitions for responsive feeding, unless this is the foundation definition as a base for this study.  There are others that show the relationship features that you address later in your discussion of findings and your conclusions.

Lines 75-89 This is an important sentence with critical information for the reader. Perhaps break this long sentence into two sentences. 

Line 83 Review the current literature about the impact of subtle pressure, even asking the child if they are still hungry. Consider including adding literature, if you feel it is appropriate.

Line 115 Please provide justification as to why qualitative exploratory design with focus groups was the best design for addressing this topic.

Line 118 Clarify “early years nutrition research context.”

Line 122-124 What was included in the initial recruitment invitation during the class time? What was the class title? 

Line 122-125 For international readers, please clarify the role of the Technical and Further Education institution in the education system. For international readers, please clarify the difference in Diploma, Certificate, and Degree

Line 124  What important differences existed between the two campuses?

Line 125 How were course offerings similar or different for those in the Diploma program and those in the Certificate program.

Line 126   What was in the informational session?

Line 129   Please clarify “course enrolment.”

Line 132 How were participants assigned to focus groups?

Line 135 Explain the qualifications of the person who facilitated the focus groups, as well as the observer.

Line 137 How long was each focus group session?

Line 144 Describe the four photographs.

Line 155 How was the codebook verified by all authors?

Line 158 How did you determine theoretical saturation where the focus group discussion sessions showed clear patterns emerging and no new information emerging?

Line 162 How many focus group sessions did each person attend? 

Line 163 Did you measure experience working in a children’s program? If yes, please include in demographics section. Did all participants have coursework that required working in a children’s program?

Line 171 Developmentally, the range of 18-31 is quite wide. Consider breaking this to 18-24 and 25-30

Line 211-404 In your review of participant comments, your coding system appears to have paralleled CFAPQ feeding practice domains. You also present many examples of concerns about communicating with families.  Why was that not selected as an emerging theme?  Please address this.

Line 408-412 Results of the study should be framed in terms of qualitative data rather than empirical data. The sample is not random, consequently one cannot generalize to the greater population. Avoid stating that trainee childcare educators are influenced in the ways described here. Please rewrite this section to explain the findings in relation to focus group standards. Explain how these results generate discussion about training and trainee perceptions.  Your exploratory data rely on people’s experiences and perceptions that provide insights for researchers and personnel preparation faculty.   

Line 410 Confirms is a definitive statement.  Rewrite to reflect the qualitative nature of your findings.

Lines 413-604 This is a well-written discussion with appropriate literature. 

Line 526 Please delete the word “therefore” which implies your study is the reason “child care educator training should aim…”

Author Response

Thank you for your comments. Please see our response attached.

Reviewer 2 Report

First of all, thank you for giving me the opportunity to review the article “Knowledge, attitudes and practices of Australian trainee childcare educators regarding their role in the feeding behaviours of young children”, submitted for publication in the International Journal of Environmental Research and Public Health. The study is interesting and the document if overall well-written. Please consider the following minor comments and suggestions as you move forward in reviewing your manuscript.

Abstract:

  • Please review the key-words used for the study. Best would be to provide MeSH terms as key-words. Keywords need to be well chosen, enabling the manuscript to be easily identified and cited.

Methods:

  • It was very interesting to see the assessment of practices using a questionnaire designed for childcare educators, and not parents, as usually seen. Authors state that the Childcare Feeding and Activity Practices Questionnaire was previously validated, however the reference is to the original validation study of this questionnaire, in the Netherlands. Since this study was performed among Australian childcare educators, could the same Dutch questionnaire be used? What are the psychometric properties of the questionnaire within your study population? Is the current sample comparable, and to what extent, to the original validation sample of the questionnaire?

Overall, the manuscript is clear, however, could be shortened, especially the Discussion section. Summarizing results where possible and finding the key messages that authors want to transmit to readers is suggested.

Author Response

(The authors gave the same response as above.)

Reviewer 3 Report

This manuscript presents an interesting insight and observation around factors that influence early childhood feeding behaviors in childcare centers. Given the critical influence of early dietary habits on overall health and wellness in childhood and later in adulthood, the authors' interest in exploring and describing potential windows of opportunity for intervention in childcare centers is commendable. As a reader in the field, I find the manuscript valuable and contributive to the field of child nutrition and health promotive dietary habits in young children. Having said this, I have few minor comments which the authors can utilize to improve the paper:

  • Although comprehensive and informative, I found the introduction section rather lengthy
  • I also noted quite a few sentences both in the introduction and other sections that could have been concise or broken into shorter sentences...I found the lengthy sentences as disrupting the flow of the reading and message the authors trying to get across.
  • consider revising some sentences that appear to have errors/ grammatical issues (e.g. line 254-257, line 352, line 475-476)
  • repeated reference to the word "placement" was made in the result and discussion sections but no description as to the nature of this placement was given in the method section as part of describing the study participants. From the context, it seems the trainee childcare educators are placed in childcare centers to gain some practical insights as part of their training...however, since several reference was made to it in the paper, the authors can explicitly describe this in the method section rather than leaving the reader to speculate what it is (e.g. how long was the placement? were the trainees allowed to participate in the routine of the center or they were there simply to observe?)
  • Authors mentioned the CFAPQ is a validated tool but didn't provide details of who developed it, for what purpose, validated for use in what context (geographic, demographic, institutional, etc. setting?)   

Author Response

(The authors gave the same response as above.)
